# Cancer Treatment Patterns and Factors Affecting Receipt of Treatment in Older Adults: Results from the ASPREE Cancer Treatment Substudy (ACTS)

**DOI:** 10.3390/cancers15041017

**Published:** 2023-02-05

**Authors:** Jaidyn Muhandiramge, Erica T. Warner, John R. Zalcberg, Andrew Haydon, Galina Polekhina, Gijsberta J. van Londen, Peter Gibbs, Wendy B. Bernstein, Jeanne Tie, Jeremy L. Millar, Victoria J. Mar, John J. McNeil, Robyn L. Woods, Suzanne G. Orchard

**Affiliations:** 1School of Public Health and Preventive Medicine, Monash University, Melbourne, VIC 3168, Australia; 2Austin Health, Heidelberg, VIC 3084, Australia; 3Clinical and Translational Epidemiology Unit, MGH Cancer Center, Massachusetts General Hospital and Harvard Medical School, Boston, MA 02114, USA; 4Department of Medical Oncology, Alfred Hospital, Melbourne, VIC 3004, Australia; 5Department of Medicine, University of Pittsburgh, Pittsburgh, PA 15260, USA; 6The Walter & Eliza Hall Institute of Medical Research, Parkville, VIC 3052, Australia; 7Walter Reed National Military Medical Center, Bethesda, MD 20814, USA; 8Department of Medical Oncology, Peter MacCallum Cancer Centre, Melbourne, VIC 3000, Australia; 9Victorian Melanoma Service, The Alfred Hospital, Melbourne, VIC 3004, Australia

**Keywords:** cancer treatment patterns, ageing, older adults, chemotherapy, radiation therapy

## Abstract

**Simple Summary:**

Planning and delivering cancer treatment in older adults with cancer is complex, given the difficulty in balancing the benefits of treatment on survival and quality of life, and the risk of harmful side effects. As a result, treatment rates in older adults with cancer tend to differ from younger patients, and there are few reports describing treatment rates in older adults. In a group of healthy older patients with cancer, we looked at rates of treatment classed as surgery, drug therapies such a chemotherapy or immunotherapy, and radiotherapy, and found that the vast majority received some form of treatment (81%), most commonly, surgery. Those with breast cancers were most likely to receive treatment (98%), in contrast to blood cancers, where patients were least likely to receive treatment (only 60% received treatment). Older patients, those residing in the US, current smokers, and those with diabetes were also less likely to receive treatment.

**Abstract:**

Introduction: Cancer treatment planning in older adults is complex and requires careful balancing of survival, quality of life benefits, and risk of treatment-related morbidity and toxicity. As a result, treatment selection in this cohort tends to differ from that for younger patients. However, there are very few studies describing cancer treatment patterns in older cohorts. Methods: We used data from the ASPirin in Reducing Events in the Elderly (ASPREE) trial and the ASPREE Cancer Treatment Substudy (ACTS) to describe cancer treatment patterns in older adults. We used a multivariate logistic regression model to identify factors affecting receipt of treatment. Results: Of 1893 eligible Australian and United States (US) participants with incident cancer, 1569 (81%) received some form of cancer treatment. Non-metastatic breast cancers most frequently received treatment (98%), while haematological malignancy received the lowest rates of treatment (60%). Factors associated with not receiving treatment were older age (OR 0.94, 95% CI 0.91–0.96), residence in the US (OR 0.34, 95% CI 0.22–0.54), smoking (OR 0.57, 95% CI 0.40–0.81), and diabetes (OR 0.56, 95% CI 0.39–0.80). After adjustment for treatment patterns in sex-specific cancers, sex did not impact receipt of treatment. Conclusions: This study is one of the first describing cancer treatment patterns and factors affecting receipt of treatment across common cancer types in older adults. We found that most older adults with cancer received some form of cancer treatment, typically surgery or systemic therapy, although this varied by factors such as cancer type, age, sex, and country of residence.

## 1. Introduction

Cancer is one of the most common diseases worldwide and a leading cause of morbidity and death [1]. The incidence of cancer in older adults is increasing, driven predominantly by increasing numbers of individuals living into old age [1]. An increase in the availability of screening and surveillance, coupled with the development of newer, more effective therapies, has led to a decrease in mortality across several cancer types [1]. In Australia, cancer survivor numbers are estimated to grow from 1.1 million in 2018 to 1.9 million in 2040 [2], and in the United States (US), an additional 5.2 million survivors are anticipated between 2019 and 2030 [3].

Cancer management in older adults is complex, with many factors determining whether treatment is offered, including patient age, frailty, and comorbidities, along with the goals of patient care [4,5]. In a cohort where life expectancy is limited irrespective of cancer, older adults with cancer are typically less likely to be offered aggressive treatment [6], sometimes leading to undertreatment of older patients who would otherwise have been deemed suitable for cancer treatment if not for their age. In those who do receive treatment, however, careful selection of treatment modality and regimen is required to avoid treatment-related toxicity and morbidity [6].

Older adults represent almost half of the global cancer population, and the proportion of older adults in the population is growing; yet, this group is frequently underrepresented in cancer treatment clinical trials [7]. Similarly, while several studies have investigated cancer treatment patterns, these are usually limited to cohorts of younger patients, often those in late middle-age [8,9]. Furthermore, these studies typically focus on a single, or few, cancer types and often do not analyse factors affecting receipt of treatment. 

We used data from the ASPirin in Reducing Events in the Elderly (ASPREE) trial, which captured cancer events and associated data, along with treatment data collected within the ASPREE Cancer Treatment Substudy (ACTS), to explore the characteristics of older persons receiving cancer treatment and the types of treatment employed. We aim to describe cancer treatment patterns in older adults diagnosed with cancer and explore the factors affecting receipt of cancer treatment.

## 2. Methods

### 2.1. Study Design

ACTS was designed as a substudy nested within the ASPREE trial. ASPREE was a randomised, double-blinded, placebo-controlled, multi-institution clinical trial investigating whether daily low dose (100 mg) aspirin prolonged disability-free survival in healthy older people. The study protocol and baseline participant characteristics of ASPREE have been previously published [10,11], along with the main trial findings [12,13,14].

### 2.2. Study Population

The ASPREE trial recruited eligible participants from 2010–2014 and collected data via annual in-person visits and regular phone calls, for a median 4.7 years. A total of 19,114 community-dwelling individuals aged ≥70 years (≥65 if a US minority) from both Australia (n = 16,703) and the US (n = 2411) were recruited mainly from general practice clinics (Australia) and research centres (US). To be eligible, participants needed to be free from cardiovascular disease, dementia, and significant physical disability. Prior history of cancer was not an exclusion criteria (19% had a past cancer history) [15] but eligible participants had to be free from a disease likely to cause death within 5 years.

ASPREE participants from both countries, with an incident cancer event during the trial, were included in ACTS. Participants with only a history of cancer pre-randomisation (i.e., no in-trial cancer) were not included. However, past cancer did not preclude participants from inclusion in ACTS if an in-trial cancer event occurred. ACTS attempted to collect cancer treatment data for the 1933 participants diagnosed with incident cancer during ASPREE.

### 2.3. ASPREE Cancer Adjudication

The details of cancer event capture and adjudication for ASPREE are described elsewhere [16]. Briefly, ASPREE captured in-trial cancer events through 6-monthly participant self-reporting and an annual review of medical records. These data included cancer type, date of diagnosis or date metastatic disease was discovered, stage, and type of event (i.e., metastatic vs. non-metastatic). Clinical documentation and investigations relating to the event were sought from general practice clinics, hospitals, specialists, and pathology provider services and compiled into a case summary for adjudication, with each case assigned to two clinical experts to confirm or refute the report based on pre-specified criteria. If the two adjudicators were not in accordance, a third adjudicator would review the case, or the case was discussed at the bimonthly adjudication committee meeting. For participants with pre-randomisation cancers, any post-randomisation cancer needed to be a new cancer or development of metastatic disease of a pre-existing cancer.

### 2.4. ACTS Data Collection

Cancer treatment data were extracted from the supporting documents sourced during ASPREE for event adjudication or from a participant’s specialist. Data were collected to define whether a participant had received cancer treatment, the cancer for which it was prescribed, and the modality of treatment. A participant’s treatment status was only listed as “No treatment” if there was definitive evidence that they had not received treatment. If treatment status for any modality was unclear, they were excluded from the analysis. Treatment categories and coding rules were developed in consultation with at least two practicing medical oncologists and are described in Appendix A. Data on treatment dose, duration, line, intent, or regimen were not collected. Treatment data were only collected for cancer events that occurred during the ASPREE time period (i.e., prior to 12 June 2017). For participants who had two cancer events for the same cancer types (e.g., a non-metastatic breast and a metastatic breast endpoint), the same treatment regime was entered for both events. If a participant developed a different type of cancer, then the treatment for each cancer type was evaluated independently.

### 2.5. Statistical Analysis

Descriptive statistics of relevant baseline characteristics and the age at cancer diagnosis for all ASPREE participants, the ACTS cohort, and “Treatment” and “No treatment” groups, are shown as simple frequencies or means and interquartile ranges. Descriptive statistics of cancer treatment patterns (rates) are shown as simple frequencies and percentages stratified by cancer type (which in turn was stratified into non-metastatic solid tumours, metastatic solid tumours, and haematological cancers), age, time from cancer diagnosis to death, and cause of death. While cancer stage data was collected during ASPREE, it was not used in this analysis due to low participant numbers following stratification by stage. The crude (unadjusted) association between each of the baseline characteristics (except age, where the age at cancer diagnosis was used) and receiving treatment was assessed either by a *t*-test or chi-square test. To explore the association between receiving treatment and the chosen clinically relevant baseline characteristics, we employed multivariate logistic regression, thus aiming to investigate the statistically independent association between receiving any treatment, or a particular type of treatment, and each variable. The reasonable assumption was made that the baseline characteristics used in the analysis were mainly unchanged at the time of cancer diagnosis, except in the case of age; here, the age at cancer diagnosis was used. The linearity assumption was assessed for continuous variables by inspecting a plot between each predictor and the logit values. Collinearity was assessed by calculating variance inflation factors and was not present between variables. As the rurality and Index of Relative Socio-economic Advantage and Disadvantage variables were only available for Australian participants, a separate model using these variables for Australian participants only was created. Sensitivity analysis was performed excluding specific ethnic groups and sex-specific cancers to investigate the impact of ethnicity and sex on receipt of treatment. All analyses were performed, and Figure 1 and Figure 2 were prepared, using software R (R Core Team, 2020).

## 3. Results

Baseline characteristics of the ACTS cohort are summarised in Table 1. Of the 1933 ASPREE participants with post-randomisation cancer, 1893 ASPREE participants were included in ACTS (median age 74.58; 56% male; 94% Non-Hispanic Caucasian; 91% Australian). Median age, race, and country of residence of ACTS participants were similar to that of the total ASPREE cohort, although a greater proportion of ACTS participants were male compared to ASPREE (44% male). Of the ACTS cohort, 1569 (83%) had received some form of cancer treatment. Nearly four-fifths consumed alcohol, while only a very small minority were current smokers. A slightly greater proportion of ACTS participants were current or former smokers compared to ASPREE participants. Similar to the ASPREE cohort, hypertension and dyslipidaemia were prevalent in the ACTS cohort (76% and 63%, respectively), while a smaller proportion had chronic kidney disease or diabetes (32% and 13% respectively). Diabetes was more prevalent in those who did not receive cancer treatment. Almost all participants (98%) were not frail upon entry to ASPREE. A flow diagram illustrating ACTS eligibility can be found in Appendix A.

The characteristics of the cancer treatment received by the ACTS cohort are outlined in Table 2. The most common type of cancer treatment was surgery (54% of those receiving treatment), followed by systemic therapy (46%), radiation therapy (29%), and regional therapy (1%) (this group includes treatments including regional chemotherapy (e.g., transarterial chemoembolisation) and regional immunotherapy (e.g., intravesical BCG). Appendix A contains the full definition for regional therapy). Usually only one major treatment modality was administered (55%), but among those receiving multimodal therapy, systemic therapy plus surgery (28% of those receiving treatment) was the most common. The most common form of systemic therapy was cytotoxic chemotherapy (62% of all systemic therapies). In those who received systemic therapy, 85% received only one type of systemic therapy (e.g., cytotoxic chemotherapy only).

Table 3 summarises the modalities of cancer treatment received by the six most common cancer types, along with the frequency of each cancer type in the ACTS cohort, and stratification of treatment by age, survival time, and cause of death. The most common cancer types diagnosed during ASPREE were blood, breast, colorectal, lung, and prostate cancers, and melanoma. The majority of these were non-metastatic (n = 937). Nearly all participants with non-metastatic breast cancer received treatment (98%), while those with non-metastatic prostate and haematological cancers were least likely to receive treatment (69% and 60% of the time, respectively). Those with metastatic breast, colorectal, and prostate cancer received treatment in similarly high proportions (88%). Generally, older participants received treatment less frequently than younger participants; those aged 70–75 years (85%) received the greatest amount of treatment, and this held across all treatment modalities. Participants with greater time from cancer diagnosis to death (i.e., survival time) more frequently received cancer treatment. Treatment data for the less common cancer types can be found in Appendix A. Treatment data stratified by sex and country can be found in Appendix A, respectively.

The impact of various baseline factors on the receipt of cancer treatment was assessed using a multivariate logistic regression model. In summary, increasing age (OR 0.94, 95% CI 0.91–0.96), residence in the US (OR 0.34, 95% CI 0.22–0.54), current smoking (OR 0.57, 95% CI 0.40–0.81), and diabetes (OR 0.56, 95% CI 0.39–0.80) reduced the likelihood of receiving any treatment. Sensitivity analysis excluding non-Caucasian participants (OR 0.37 95% CI 0.23–0.61) and African American participants (OR 0.40, 95% CI 0.26–0.62) attenuated the impact of US residence but did not remove it entirely. Female sex was associated with receipt of treatment (OR 1.63, 95% CI 1.24–2.17), although on sensitivity analysis excluding sex-specific cancers, this association was no longer present (OR 1.05, 95% CI 0.77–1.44). The results of the models, including those analysing receipt of systemic therapy, radiation therapy, and surgery, are detailed in Figure 1 and Figure 2.

## 4. Discussion

The ASPREE Cancer Treatment Substudy (ACTS) adds valuable information to the literature regarding cancer treatment patterns in older adults with various types of cancer. In our cohort, most older but otherwise reasonably healthy adults with cancer received some form of cancer treatment, with roughly half receiving at least one of systemic therapy or surgery and younger participants more frequently receiving treatment. Slightly more female participants received treatment than males (86% vs. 80%), while more Australian participants received treatment than US participants (84% vs. 71%). Treatment rates generally increased with increasing survival time, although this is not surprising given that those with advanced disease and short life expectancies are unlikely to be offered treatment, particularly in cases where systemic therapy or surgery requires a reasonable functional baseline [17]. 

Our treatment pattern data generally aligns with the cancer data collected by the National Cancer Registration and Analysis Service in England, where 45% received surgery and 27% received radiation therapy. While the service’s 2020 report did not look at systemic therapy overall, it states that 28% received cytotoxic chemotherapy, in line with our cohort [18]. Notably, the English data represents all patients with cancer, not just older adults. While some Australian cancer data exist in the form of National Cancer Control Indicators published by Cancer Australia [19], cancer treatment patterns are not widely available. Similarly, data from the US on overall cancer treatment patterns is not easily accessible, although Miller et al. present a detailed analysis of site-specific treatment patterns using Surveillance, Epidemiology, and End Results (SEER) data [20]. While overall treatment patterns provide a broad overview of how cancer in older adults is managed, disease-specific stratification is essential given the heterogeneity in treatments offered for various cancer types. In general, treatment rates for the six most common cancers in our study did not differ greatly from the existing literature.

Participants with haematological malignancies received treatment in slightly greater rates than older cohorts with acute leukaemia in the US [21,22], but in lower rates than US patients with slower growing haematological malignancies [23]. As ASPREE grouped all types of haematological malignancy together, the heterogeneity of cancer aggressiveness, treatments offered, and stage may provide an explanation for this variation. Participants with haematological malignancy also received some of the lowest rates of treatment overall (60%), possibly due to a predominance of classically non-aggressive subtypes (e.g., chronic lymphocytic leukaemia), where a ‘watchful waiting’ approach is often employed [20,24].

For breast cancer, our data resemble published data from both Australia and the US, where most patients with non-metastatic breast cancer undergo surgery [20,25,26]. Rates of surgical treatment were greatest in this group, likely due to the use of relatively low-risk breast-conserving procedures. Our cohort received slightly less radiation therapy than that reported in a younger Australian cohort (median age 61 versus 74.6 years), where 63% received radiation therapy [25]. Rates of radiation therapy were still the highest for the six commonest cancers, an expected finding given the widespread use of adjuvant radiation in localised or regional breast cancers [20]. In those with metastatic disease, rates of chemotherapy (31%) and hormonal therapy (63%) resemble the rates of large European and North American cohorts [9]. Here, hormone receptor status data can add nuance to treatment patterns, although that data were not available for our analysis.

We demonstrated similar treatment patterns for prostate cancer compared to the published literature, where roughly one-third undergo surgery and a slightly higher proportion receive radiation [27], although younger Australian men are more likely to undergo radical prostatectomy (47% as described by Wang et al.) [28]. Radiation was the most common treatment (39%) for early-stage prostate cancer, an expected finding given the similar efficacy [29] and relative safety of radiation therapy versus surgery, particularly in our older cohort. ACTS participants with metastatic prostate cancer received hormonal therapy (88%) in a similar proportion to an older cohort of American patients [8]. Notably, of the commonest solid tumours, non-metastatic prostate cancer demonstrated the highest rate of ‘no treatment’, likely due to the use of ‘watchful waiting’ in older men with prostate cancer to prevent overtreatment [28].

For patients with non-metastatic colon cancer, Beckmann et al. demonstrated similarly high rates (83%) of surgery in an Australian cohort [30] when compared to our cohort (89%). Comparison of radiation therapy is difficult given that we did not differentiate between colon and rectal cancers in our study; nearly one-third of patients with rectal cancer in the aforementioned study received radiation therapy, compared to very few of those with colon cancer [30]. Rates of systemic therapy, however, were roughly similar to data from both Australia and the US [20,30], with usage increasing with increasing stage. Of all cancer types, the use of targeted therapies was greatest in metastatic colorectal cancers, where EGFR (epidermal growth factor receptor) inhibitors (e.g., cetuximab) and anti-angiogenic agents (e.g., bevacizumab) are commonly used [30]. 

Roughly half of ACTS participants with early-stage lung cancer received surgery, echoing US data [20]. Notably, we did not differentiate non-small cell lung cancer and small cell lung cancer, an important delineation given that the latter is often metastatic on presentation and is therefore rarely resected. A frequency of cytotoxic chemotherapy use in patients with metastatic lung cancer receiving systemic therapy follows previously published data from Australia [31], the US [20], and the United Kingdom [32]. Our older cohort, however, received less systemic therapy (50%) for advanced lung cancer when compared to younger cohorts; for example, Ngo et al. reported receipt of systemic therapy in 76% and 65% of patients aged <60 and 60–69 years, respectively [31], while only six ACTS participants (8%) received immunotherapy compared to 12% of similar patients in the US [20]. It should be noted that immunotherapy was not routinely used during the ASPREE time period (pre-2017). Targeted therapy was as popular for advanced lung cancer in ACTS as immunotherapy; EGFR inhibitors and anti-angiogenic agents are similarly increasingly used in lung cancer [33].

Most melanomas were treated with resection in both our cohort and worldwide [20]. A small number of patients with non-metastatic melanoma also received immunotherapy or radiation therapy. Certainly, in the ‘real world’, stage III melanomas may be offered adjuvant targeted therapy [34] or immunotherapy [35], although these treatments were not available during the ASPREE timeframe and are less frequently used in the older population. Seventeen percent of patients with metastatic melanomas received targeted therapy; in recent years, BRAF/MEK inhibitors have shown survival benefits in BRAF-mutated disease [36]. Similarly, immunotherapy is a common modality used in metastatic melanoma, along with radiation [20]; roughly half of our cohort received one of the two.

We used logistic regression models to determine which factors impacted receipt of cancer treatment. In line with studies worldwide [18,19,20], participants were less likely to receive any treatment, systemic therapy, radiotherapy, or surgery as they aged. In the case of systemic therapy, performance status, often measured using the Eastern Cooperative Oncology Group performance status (ECOG) [17], is a key consideration in whether treatment is offered to an older patient. Similarly, pre-operative frailty and comorbidities (both of which are common in older age) are associated with post-operative morbidity and mortality and can preclude older patients from undergoing surgery [37]. While our primary model found that female participants were more likely to receive treatment, a finding that aligns with national Australian data [19], this association was no longer present when sex-specific cancers were excluded. The sex difference is therefore likely driven by differing treatment patterns between the most common sex-specific cancer types; in ACTS participants with non-metastatic breast cancer, nearly all received some form of treatment, compared to only 69% of participants with non-metastatic prostate cancer. In non-sex specific cancers, we found that treatment rates were roughly similar, a finding echoed by English data [18]. US-based participants were significantly less likely to receive most treatments compared to Australian participants. To our knowledge, there have been no head-to-head comparisons of access to cancer treatment between these two countries. Sensitivity analysis showed that this difference was partly driven by the preponderance of ethnic minorities in the US ASPREE cohort. Beyond this, however, the difference may be explained by the markedly different health systems in Australia and the US. In Australia, financial capacity is rarely a barrier to treatment, whereas high out-of-pocket expenses in the US can limit access. Notably, the small sample size of the US cancer cohort may bias the odds ratio, overestimating the effect of country of residence; therefore, the magnitude of this association should be interpreted with caution.

Several modifiable risk factors also impacted receipt of treatment. Smokers were nearly twice as likely to not receive any treatment or systemic therapy. Smoking status is not commonly investigated in similar studies of treatment patterns, although Ngo et al. reported no significant variation in receipt of treatment between current, former, and non-smokers in patients with lung cancer receiving systemic therapy [31]. However, it is well-established that smoking can reduce the effectiveness of chemotherapy for lung cancer [38], increase symptom burden in those receiving chemotherapy and radiation therapy [39], and increase surgical risk [40], possibly impeding such patients from being offered treatment. Interestingly, current smokers were more likely to receive radiation therapy. This may be driven by smokers being less likely to receive systemic therapy or surgery due to the aforementioned risks, and instead being offered lower risk, local radiation therapy, although there is little published literature to support this hypothesis. Comorbidities may also be associated with decreased rates of treatment [4,18,41], although the evidence for whether comorbidities themselves impact the receipt and efficacy of treatment is equivocal [42,43]. Instead, it may be that decreased performance status and/or poor baseline function secondary to comorbidities plays a greater role. Nonetheless, ACTS participants with diabetes were less likely to receive any cancer treatment [44]. Diabetes can increase postoperative risk [45], although this was not reflected in surgical treatment data in our cohort. Chronic kidney disease in our cohort was not statistically significantly associated with varying rates of any treatment. While chronic kidney disease may not impact suitability for radiation therapy or surgery, it can increase the risk of toxicity and adverse reactions to systemic therapy [46].

Notably, we did not find a statistically significant association between race, rurality, or socioeconomic status, and receipt of treatment, except for those of Asian background and receipt of radiation therapy. All of these factors have been previously found to be associated with varying rates of cancer treatment [4,19,47], although in Australia, remoteness and socioeconomic status do not appear to impact receipt of treatment [19]. Note that our data do not provide information on whether these factors affect ease of access to cancer treatment or timeliness of treatment. Nonetheless, our findings may be explained by variation between cohorts, or that our sample sizes for each stratum were relatively small. Notably, while our regression models did not reveal significant associations between these factors, the descriptive data did highlight some differences. For example, 83% of Caucasian participants received treatment compared with 75% of African American participants. Similarly, we saw that treatment rates decreased with increasing rurality, although the majority of those living in inner and outer regional Australia still received treatment (83 and 79%, respectively). Furthermore, the impact of these factors is likely to vary between countries, depending on health systems, level of reimbursement, and location-based access to treatment. Alternatively, it has been suggested that these differences may be explained by different cancer types or characteristics seen in different races or socioeconomic groups [18]. We also found no significant impact of frailty on receipt of cancer treatment, although due to ASPREE’s strict inclusion criteria, few participants were frail at enrolment. Furthermore, frailty was only measured at baseline in ASPREE; it is likely that over the course of the study, more participants developed frailty. Similarly, we found no statistically significant impact of BMI on receipt of treatment, despite previous studies describing that high (but not excessively high) BMI can improve outcomes for both systemic therapy [31] and surgery [48]. For both frailty and BMI, however, the point estimates for odds ratios were less and greater than 1.00, respectively, and may therefore partly reflect a need for a larger sample. 

This analysis has several strengths. Firstly, the large cohort of the ASPREE trial allowed for a relatively large number of cancer events to be captured with a reasonable spread of cancer types and stages, along with a reasonably representative sample. However, the limited sample size in the US and the modest number of participants within each cancer type limit our ability to investigate heterogeneity in treatment patterns across cancer types and countries. The adjudication of cancer cases allowed for accurate classification of cancer events. The in-depth baseline data captured by ASPREE was also a significant strength and allowed us to examine the relationship of various participant characteristics and receipt of treatment. Finally, cancer treatment data were available for a significant proportion (98%) of the ACTS cohort, allowing for a relatively robust dataset. The rigorous inclusion criteria for ASPREE requiring participants to be ‘healthy’ and devoid of a condition likely to cause death within five years, however, may be a limitation in our analysis. These participants are typically more likely to receive treatment due to greater baseline functional status, thereby overestimating treatment rates. A lack of specific treatment data also prevents more detailed analysis of treatment patterns. In particular, the use of broad treatment categories without data on specific treatment type, dose, line, or duration limited our ability to provide information on factors affecting receipt of treatment that could be used to impact clinical practice. We also did not collect data on treatment intent; as a result, we were unable to differentiate between rates of curative versus palliative treatment.

Despite cancer treatment typically having a positive impact on cancer-related mortality and recurrence, both cancer and cancer treatment are thought to have the potential to accelerate adverse ageing outcomes in cancer survivors [49,50]. Therefore, the ACTS substudy data will be used in subsequent analyses to explore long-term outcomes of older patients with cancer, including the impact of cancer and cancer treatment on cognitive decline and dementia, cardiovascular disease, and functional decline.

## 5. Conclusions

This study is one of the first to describe cancer treatment patterns and factors associated with receipt of treatment across several cancer types in older adults in Australia and the US. In our cohort, most older adults with cancer received some form of cancer treatment, typically only one type, and most commonly, surgery. Younger and female participants more frequently received cancer treatment than older participants, although the sex association was predominantly driven by differing treatment rates for sex-specific cancers. Residence in the US, current smoking status, and diabetes reduced the likelihood of receiving any treatment.

## Figures and Tables

**Figure 1 cancers-15-01017-f001:**
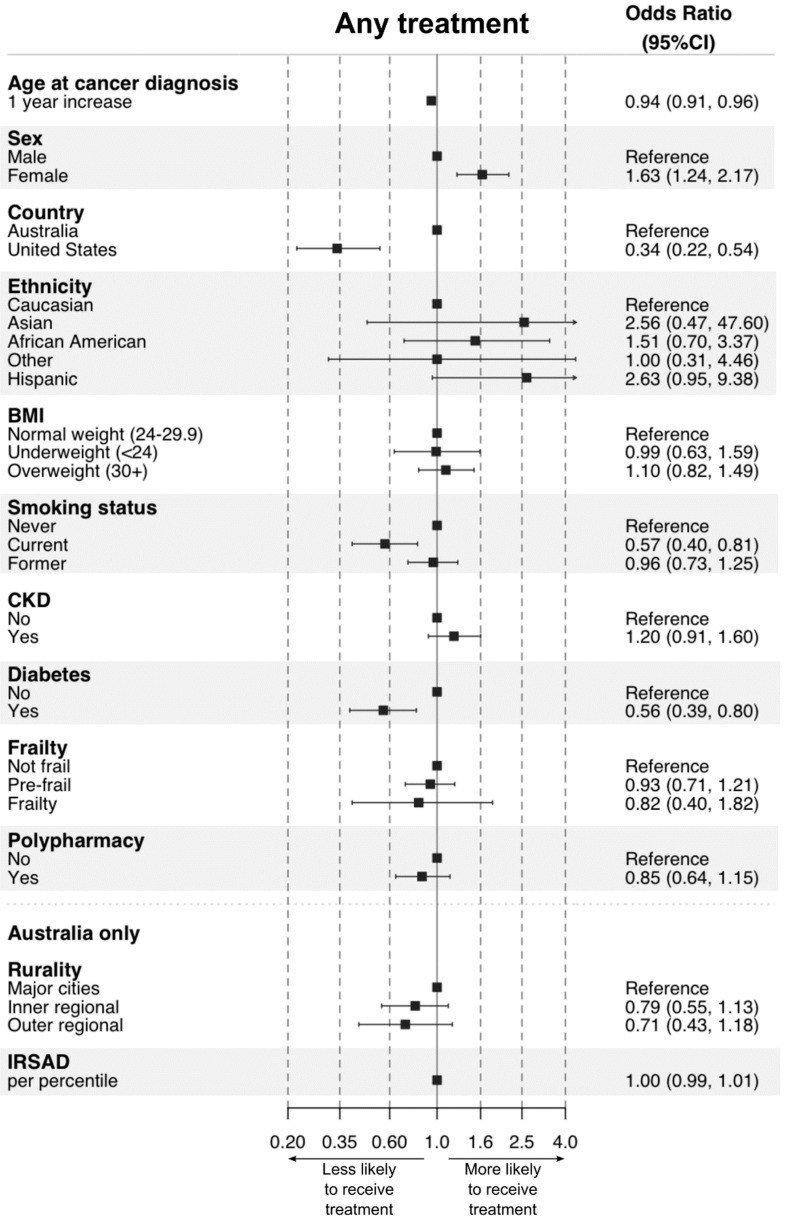
Associations between demographic factors, health behaviours, and chronic conditions and receipt of any cancer treatment for incident post-randomisation cancers during ASPREE.

**Figure 2 cancers-15-01017-f002:**
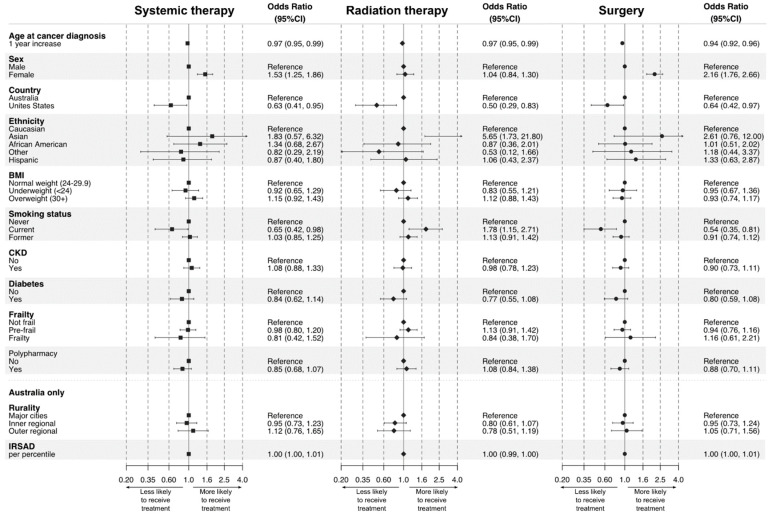
Associations between demographic factors, health behaviours, and chronic conditions and receipt of systemic therapy, radiation therapy, and surgery for incident post-randomisation cancers during ASPREE (Note that the models for each modality are independent of one another, and therefore ORs cannot be compared between modalities).

**Table 1 cancers-15-01017-t001:** Participant demographics and characteristics of the total cohort, as well as stratification by incident (first) post-randomisation cancer diagnosis and receipt of cancer treatment.

	Total ASPREE Cohort n = 19,114 (% of Column)	ACTS Cohort			*p* Value (No Treatment vs. Treatment)
		Total n = 1893 (% of Column)	No Cancer Treatment n = 324 (% of Column; % of Row)	Cancer Treatment n = 1569 (% of Row; % of Column)	
Age at cancer diagnosis [years; median (Q1–Q3)]	N/A	77.3 (74.6–81.4)	78.87 (75.1–84.0)	77.14 (74.5–80.9)	<0.001
**Sex**					
Male	8332 (44%)	1053 (56%)	210 (65%, 20%)	843 (54%, 80%)	<0.001
Female	10,782 (56%)	840 (44%)	114 (35%, 14%)	726 (46%, 86%)	
**Race**					
Non-Hispanic Caucasian	17,449 (91%)	1770 (94%)	303 (93%, 17%)	1469 (94%, 83%)	0.359
Non-Hispanic Asian	164 (1%)	13 (1%)	1 (<1%, 8%)	12 (1%, 92%)	
Non-Hispanic African American	901 (5%)	56 (3%)	14 (4%, 25%)	42 (3%, 75%)	
Non-Hispanic Other	111 (<1%)	18 (1%)	4 (1%, 22%)	14 (1%, 78%)	
Hispanic	488 (3%)	35 (2%)	4 (1%, 11%)	31 (2%, 89%)	
**Country**					
Australia	16,703 (87%)	1718 (91%)	274 (85%, 16%)	1444 (92%, 84%)	<0.001
United States	2411 (13%)	175 (9%)	50 (15%, 29%)	125 (8%, 71%)	
**Rurality ^a^**					
Major cities	8729 (52%)	947 (50%)	136 (50%, 14%)	811 (56%, 86%)	0.084
Inner regional	5976 (36%)	587 (31%)	100 (37%, 17%)	487 (34%, 83%)	
Outer regional	1947 (12%)	180 (9%)	37 (14%, 21%)	143 (10%, 79%)	
IRSAD percentile ^b^	58 (31–83)	59 (34–85)	60 (35–85)	57 (30–82)	0.088
**Education**					
≤12 years of education	10,955 (57%)	1088 (57%)	194 (60%, 18%)	894 (57%, 82%)	0.403
13–15 years of education	3255 (17%)	345 (18%)	50 (15%, 14%)	295 (19%, 86%)	
≥16 years of education	4903 (26%)	460 (24%)	80 (25%, 17%)	380 (24%, 83%)	
**BMI ^c^**					
Underweight (<24)	2194 (12%)	182 (10%)	32 (10%, 18%)	150 (10%, 82%)	0.771
Normal weight (24–30)	11,224 (59%)	1150 (61%)	201 (62%, 17%)	949 (61%, 83%)	
Overweight (≥30)	5607 (30%)	553 (29%)	90 (28%, 16%)	463 (30%, 84%)	
**Alcohol use**					
Current	14,642 (77%)	1492 (79%)	251 (78%, 17%)	1241 (79%, 83%)	0.681
Former	1136 (6%)	125 (6%)	25 (8%, 20%)	100 (6%, 80%)	
Never	3336 (18%)	276 (15%)	48 (15%, 17%)	228 (15%, 83%)	
**Smoking status**					
Current	735 (4%)	120 (6%)	29 (9%, 23%)	91 (6%, 77%)	0.081
Former	7799 (41%)	850 (45%)	147 (45%, 17%)	703 (45%, 83%)	
Never	10,580 (55%)	923 (49%)	148 (46%, 16%)	775 (49%, 84%)	
**Chronic disease**					
Chronic kidney disease	4740 (25%)	573 (30%)	98 (32%, 17%)	475 (32%, 83%)	1.000
Diabetes	2045 (11%)	242 (13%)	65 (20%, 27%)	177 (11%, 73%)	<0.001
Dyslipidaemia	12,467 (65%)	1195 (63%)	200 (61%, 17%)	995 (63%, 83%)	0.522
Hypertension	14,195 (74%)	1437 (76%)	252 (77%, 18%)	1185 (76%, 82%)	0.542
**Frailty ^d^**					
Not frail	11,246 (59%)	1066 (56%)	163 (50%, 15%)	903 (58%, 85%)	0.033
Pre-frailty	7447 (39%)	779 (41%)	149 (46%, 19%)	630 (40%, 81%)	
Frailty	421 (2%)	48 (3%)	12 (4%, 25%)	36 (2%, 75%)	
Polypharmacy	5088 (27%)	506 (27%)	101 (31%, 20%)	405 (26%, 80%)	0.064
Aspirin intervention	9589 (50%)	930 (49%)	163 (50%, 18%)	767 (49%, 82%)	0.760

Abbreviations: BMI = Body Mass Index, IRSAD = Index of Relative Socio-economic Advantage and Disadvantage. NB: Some percentages may not add to 100% where there are participants with missing data. ^a^ Australian participants only. ^b^ A summary measure of economic and social conditions within an area. A low score corresponds to greater disadvantage. Here, we present percentiles for ease of interpretation. ^c^ BMI as per recommendations for older adults (>65 years). ^d^ As classified using the Fried Frailty Index (presence of at least three of the following; unintentional weight loss, poor handgrip strength, self-reported exhaustion, slow gait speed, and low physical activity).

**Table 2 cancers-15-01017-t002:** Characteristics of the incident cancer diagnosed following randomisation and the treatment in participants who received a post-randomisation cancer diagnosis during ASPREE.

	Cancer Treatment Cohort (n = 1893)
**No cancer treatment ^a^**	324 (17%)
**Any cancer treatment ^a^**	1569 (83%)
Systemic therapy	869 (46%)
Cytotoxic chemotherapy	537 (28%)
Hormonal therapy	351 (19%)
Targeted therapy	85 (5%)
Immunotherapy	31 (2%)
Radiation therapy	544 (29%)
Surgery	1029 (54%)
Regional therapy	16 (1%)
**Combination therapy** ** ^b^ **	
Systemic therapy and surgery	435 (28%)
Systemic therapy and radiation therapy	368 (23%)
Radiation therapy and surgery	266 (17%)
Radiation therapy, systemic therapy, and surgery	188 (12%)
**Number of major treatment modalities ^b,c^**	
Only one modality	868 (55%)
Two modalities	505 (36%)
Three modalities	188 (12%)
**Number of types of systemic therapy ^d^**	
Only one type	741 (85%)
Two types	121 (14%)
Three or more types	7 (1%)

Percentages in some subgroups may not total to 100% as some participants received multiple modalities of cancer treatment. ^a^ Percentage of total ACTS cohort (n = 1893). ^b^ Percentage of participants who received any cancer treatment (n = 1569). ^c^ The eight missing participants in this group received “Regional therapy” and were not counted as having received a “Major treatment modality”. ^d^ Percentage of participants who received systemic therapies (n = 869).

**Table 3 cancers-15-01017-t003:** Cancer treatment modalities received for incident post-randomisation cancers during ASPREE, stratified by most common cancer types and metastatic status, and cause of death and time from diagnosis to death.

		Systemic Therapy						
	Any Treatment(n = 1569)	Any Systemic Therapy (n = 869)	Chemotherapy (n = 537)	Hormonal Therapy (n = 351)	Targeted Therapy (n = 85)	Immuno-Therapy (n = 31)	Radiation Therapy (n = 544)	Surgery (n = 1029)
**Non-metastatic solid tumours (n = 937)**								
Breast (n = 214)	210 (98%)	171 (80%)	49 (23%)	149 (70%)	13 (6%)	-	104 (49%)	206 (96%)
Colon/rectum (n = 210)	191 (91%)	71 (34%)	71 (34%)	-	4 (2%)	-	20 (10%)	187 (89%)
Lung (n = 78)	65 (83%)	20 (26%)	18 (23%)	-	1 (1%)	2 (3%)	33 (42%)	38 (49%)
Melanoma (n = 160)	151 (94%)	5 (3%)	-	-	-	5 (3%)	5 (3%)	146 (91%)
Prostate (n = 275)	191 (69%)	98 (36%)	5 (2%)	96 (35%)	-	-	106 (39%)	83 (30%)
**Metastatic solid tumours (n = 297)**								
Breast (n = 32)	28 (88%)	26 (81%)	10 (31%)	20 (63%)	4 (13%)	-	15 (47%)	8 (25%)
Colon/rectum (n = 57)	50 (88%)	42 (74%)	42 (74%)	-	20 (35%)	-	11 (19%)	35 (61%)
Lung (n = 78)	59 (76%)	39 (50%)	36 (46%)	-	6 (8%)	6 (8%)	40 (51%)	15 (19%)
Melanoma (n = 30)	23 (77%)	15 (50%)	2 (7%)	-	5 (17%)	11 (37%)	10 (33%)	16 (53%)
Prostate (n = 100)	88 (88%)	86 (86%)	25 (25%)	83 (86%)	-	-	35 (35%)	18 (18%)
**Haematological cancers (n = 187)**	112 (60%)	102 (55%)	97 (52%)	-	19 (10%)	2 (1%)	16 (9%)	10 (5%)
**Age**								
65–69 (n = 33)	22 (67%)	12 (36%)	8 (24%)	3 (9%)	1 (3%)	-	2 (6%)	16 (48%)
70–75 (n = 992)	839 (85%)	472 (48%)	300 (30%)	190 (19%)	46 (5%)	16 (2%)	301 (30%)	586 (59%)
76–80 (n = 544)	444 (82%)	252 (46%)	157 (29%)	101 (19%)	25 (5%)	12 (2%)	159 (29%)	278 (51%)
81–85 (n = 272)	201 (74%)	102 (38%)	58 (21%)	41 (15%)	10 (4%)	3 (1%)	64 (24%)	118 (43%)
85+ (n = 92)	63 (68%)	31 (34%)	14 (15%)	16 (17%)	3 (3%)	-	18 (20%)	31 (34%)
**Time from cancer diagnosis to death**								
<30 days (n = 84)	28 (33%)	13 (15%)	10 (12%)	4 (5%)	-	-	6 (7%)	14 (17%)
30–89 days (n = 82)	51 (62%)	24 (29%)	20 (24%)	2 (2%)	5 (6%)	1 (1%)	22 (27%)	14 (17%)
90–365 days (n = 193)	158 (82%)	104 (54%)	91 (47%)	9 (5%)	10 (5%)	3 (2%)	68 (35%)	75 (39%)
1–3 years (n = 143)	126 (88%)	100 (70%)	88 (62%)	15 (10%)	19 (13%)	5 (3%)	56 (39%)	70 (49%)
3+ years (n = 44)	37 (84%)	24 (55%)	15 (34%)	12 (27%)	2 (5%)	2 (5%)	12 (27%)	20 (45%)
**Cause of death**								
Cancer-related (n = 498)	367 (74%)	244 (49%)	208 (42%)	35 (7%)	35 (7%)	11 (2%)	156 (31%)	176 (35%)
Cardiovascular (n = 17)	11 (65%)	8 (47%)	6 (35%)	3 (18%)	-	-	3 (18%)	4 (24%)
Major haemorrhage (n = 6)	5 (83%)	3 (50%)	2 (33%)	2 (33%)	1 (17%)	-	1 (17%)	2 (33%)

Percentages represent the proportion of participants who received cancer treatment relative to the first column. For example, 98% of participants with breast cancers received any cancer treatment. “-” indicates that no one (0 participants) in this group received a particular treatment modality.

## Data Availability

Data are available from the corresponding author upon reasonable request and approval of the ASPREE Steering Committee.

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
