# Peer review of "Cancer Treatment Patterns and Factors Affecting Receipt of Treatment in Older Adults: Results from the ASPREE Cancer Treatment Substudy (ACTS)"

_cancers, 2023, doi:10.3390/cancers15041017_

Round 1
Reviewer 1 Report
Enormous job,but not original
Author Response
Thank you for the feedback. We agree that our research is broadly similar to other studies describing treatment patterns in patients with cancer. However, our research is original in that it describes a cohort of older patients, explores treatment patterns across several cancer types, and analyses various factors affecting receipt of cancer treatment. Evidence concerning treatment patterns and strategies in older patients is still inadequate, since most data are from clinical trials with younger patients, in which older patients have been often neglected and underrepresented. We believe that all of these factors separate our analysis from other similar studies describing cancer treatment patterns, thereby addressing the knowledge gap in older patients, an increasingly growing sector of the population, and hence, adds value to the literature. In response to the need to improve the research design, description of methods and results, and linkage of conclusion with results, we have reworked the methods, results and conclusion sections.
Reviewer 2 Report
The authors present the results of a cancer cohort within the ASPREE study. That study focuses on older patients and includes several relevant aspects for this population: e.g. frailty, comorbidity. Their conclusions have a couple of interesting aspects: 1) no gender difference in treatment patterns when sex-specific cancers are excluded; 2) Australian patients were more likely to receive treatment than US patients. Overall the paper is well written and the discussion is thorough.
The strengths of the study are: the inclusion of patients from two countries with similar prospectively gathered information and periodic assessments (per ASPREE protocol), which allows more uniformity than cancer registries. A good analysis of several potential confounders.
The limitations are that the treatment information is only qualitative with broad categories (e.g. chemotherapy without specifying types or lines of treatments) and limited ability to detect adequacies in treatment planning. The latter is however well addressed in the discussion.
The first sentence on top of page 4 is a bit unclear: "the most common for of systemic therapy was cytotoxic chemotherapy...with usually one therapy administered..." Do the authors mean chemotherapy was the only modality given? Or only one line of treatment? Or only one drug? It seems from later in the text that it refers to only one modality, but could be clarified here.
Author Response
The authors present the results of a cancer cohort within the ASPREE study. That study focuses on older patients and includes several relevant aspects for this population: e.g. frailty, comorbidity. Their conclusions have a couple of interesting aspects: 1) no gender difference in treatment patterns when sex-specific cancers are excluded; 2) Australian patients were more likely to receive treatment than US patients. Overall the paper is well written and the discussion is thorough.
The strengths of the study are: the inclusion of patients from two countries with similar prospectively gathered information and periodic assessments (per ASPREE protocol), which allows more uniformity than cancer registries. A good analysis of several potential confounders.
The limitations are that the treatment information is only qualitative with broad categories (e.g. chemotherapy without specifying types or lines of treatments) and limited ability to detect adequacies in treatment planning. The latter is however well addressed in the discussion.
Thank you for the kind words and feedback.
We certainly agree that our study is limited by a lack of detailed treatment data. We have added more detail on this limitation in the limitations section of the discussion.
The first sentence on top of page 4 is a bit unclear: "the most common for of systemic therapy was cytotoxic chemotherapy...with usually one therapy administered..." Do the authors mean chemotherapy was the only modality given? Or only one line of treatment? Or only one drug? It seems from later in the text that it refers to only one modality, but could be clarified here.
You are correct in saying that we are referring to only one modality. This has been edited within the manuscript to provide better clarity, and shown below:
The most common form of systemic therapy was cytotoxic chemotherapy (62% of all systemic therapies). In those who received systemic therapy, 85% received only one type of systemic therapy (e.g. cytotoxic chemotherapy only).
Reviewer 3 Report
In this manuscript, the authors provide data on treatment pattern of cancer for patients within the ASPREE/ASCT trial and their association with different factors.
Overall, the manuscript is well written und provides interesting results; thus, I would like to ask only for minor changes:
1) Please change the term "elderly" into "older adults" thoughout the manuscript.
2) In the introduction, it sounds as if overtreatment is the only problem for older adults with cancer. Please add and describe ageism and undertreatment as other potential issues.
3) Methods: 2.5 statistical analysis (p. 3): please describe your statistical analysis in greater details and make sure that all applied methods are described; e.g., the sensitivity analysis is not even mentioned.
4) Methods: please add how frailty was defined.
5) Figure 1 and Figure 2 are different in style. Please adjust. Which program was used to create these figures?
Author Response
In this manuscript, the authors provide data on treatment pattern of cancer for patients within the ASPREE/ASCT trial and their association with different factors.
Overall, the manuscript is well written und provides interesting results; thus, I would like to ask only for minor changes:
1) Please change the term "elderly" into "older adults" thoughout the manuscript.
Thank you for the kind words and feedback.
We have removed the term “elderly” from the manuscript and replaced it with the more appropriate ‘older adults’.
2) In the introduction, it sounds as if overtreatment is the only problem for older adults with cancer. Please add and describe ageism and undertreatment as other potential issues.
We have added the risk of undertreatment into paragraph 2 of the introduction.
3) Methods: 2.5 statistical analysis (p. 3): please describe your statistical analysis in greater details and make sure that all applied methods are described; e.g., the sensitivity analysis is not even mentioned.
We apologise for the lack of detail here. We have added more detail to our statistical methods and rewritten the entire section (2.5) such that the analysis could be replicated if necessary, including details on the sensitivity analysis.
4) Methods: please add how frailty was defined.
Frailty was defined using the Fried Frailty Index. The specific criteria used in this definition have been added to the footnote of Table 1.
5) Figure 1 and Figure 2 are different in style. Please adjust. Which program was used to create these figures?
These have been adjusted. The figures were created in R (R Core Team, 2020). We have now specified this in the “Statistical analysis” section.
Reviewer 4 Report
The goals of the study is intriguing, as the undertreatment of older canceer patients is a poorly investigated problem.
However, the present analyses is focused on fit patients, as eligible participants had to be free from a disease likely to cause death within 5 years.
This limit is not outlined in the discussion.
Moreover, the collected data about cancer are too scanty. Cancer stage, burden of disease and cancer biology (hystological grade, etc) are essential to investigate the determinant of cancer treatment choice, in addition to geriatric characteristics. A older patient who was diagnosed with a small, G1, poorly aggressive breast cancer could be correctly spared from adjuvant hormonal treatment, and this does not represent and undertreatment.
Without such details, the present study is just a "photograph" and its results do not add new, useful informations.
Author Response
The goals of the study is intriguing, as the undertreatment of older cancer patients is a poorly investigated problem. However, the present analyses is focused on fit patients, as eligible participants had to be free from a disease likely to cause death within 5 years. This limit is not outlined in the discussion.
We agree that our study is limited by its focus on healthy participants, potentially creating a healthy survivor bias. As outlined in the second-to-last paragraph of the discussion, this likely results in an overestimation of treatment rates in this population. This section in the discussion has been expanded with more detail, and highlighted in the revised manuscript.
“The rigorous inclusion criteria for ASPREE requiring participants to be ‘healthy’, and devoid of a condition likely to cause death within 5 years, however, may be a limitation in our analysis. These participants are typically more likely to receive treatment due to greater baseline functional status, thereby overestimating treatment rates.”
Moreover, the collected data about cancer are too scanty. Cancer stage, burden of disease and cancer biology (histological grade, etc) are essential to investigate the determinant of cancer treatment choice, in addition to geriatric characteristics. A older patient who was diagnosed with a small, G1, poorly aggressive breast cancer could be correctly spared from adjuvant hormonal treatment, and this does not represent and undertreatment.
Without such details, the present study is just a "photograph" and its results do not add new, useful information.
We also acknowledge this limitation in our study and have described this in greater detail in the limitations section of our revised manuscript. However, we believe that our data still adds value to the discussion surrounding cancer treatment in older adults. While we do not have detailed data on cancer characteristics, the division of each cancer type into metastatic and non-metastatic subgroups allows us to differentiate somewhat between cancers of varying aggressiveness (ASPREE did collect some cancer stage data but once the data are stratified into cancer type then stage, the numbers in the treatment patterns table become quite low). Thus, while we do only provide a ‘snapshot’ of cancer treatment patterns, it does give a reasonable overview of which cancers receive certain types of treatment (and which do not).
Round 2
Reviewer 1 Report
Again,even after the corrections the manuscript does not add anything new
Reviewer 4 Report
I appreciated the revision to discussion, as regard the limits of the trial.